

# Impacts of Source Regions and Atmospheric Transport on Physical Properties of Black Carbon and Tracer Ratios over the Yellow Sea: Evidence from Multi-Seasonal Airborne Observations

Naki Yu[1], Hee-Jung Yoo[2], Sangmin Oh[2], Yongjoo Choi[3], Sunran Lee[2], Sumin Kim[2], Saehee Lim[1,4,*]

[1]Department of Environmental & IT Engineering, Chungnam National University, Daejeon, 34134, Republic of Korea
[2]Global Atmospheric Watch and Research Division, National Institute of Meteorological Sciences, Jeju, 63568, Republic of Korea
[3]Department of Environmental Science, Hankuk University of Foreign Studies, Yongin, 17035, Republic of Korea
[4] Department of Environmental Engineering, Chungnam National University, Daejeon, 34134, Republic of Korea

*Correspondence to*: Saehee Lim (saehee.lim@cnu.ac.kr)

**Abstract.** Both size and mixing state of Black carbon (BC) are critical climate-relevant physical parameters. It remains a challenge for ambient measurements to characterize their variability across different atmospheric conditions particularly in outflow regions. To investigate how BC's physical properties are determined in source regions and altered during transport, we conducted 23 flight measurements of BC, CO, and $CO_2$ over the Yellow Sea from 2021 to 2022. The refractive BC mass concentration ($M_{rBC}$) varied by up to two orders of magnitude between near sea surface and around 5 km above sea level, and Planetary Boundary Layer height-dependency of $M_{rBC}$ was stronger in winter than in spring. Smallest rBC's mass median diameter MMD (163.4 nm) observed in South Korea-sourced air indicated fresh urban emissions, whereas larger MMD, enhanced internal mixing, and higher rBC/CO slopes were exhibited in the air masses from North Korea and China, reflecting additional emissions from biomass and coal combustion. Both MMD and internal mixing tended to decrease with altitude, highlighting the wet scavenging effect during particle transport. When accumulated precipitation exceeded 1 mm, $M_{rBC}$ decreased by more than 50%, with moderate reductions in MMD and internal mixing. As a result, overall BC transport efficiency declined to 1/e within 5.5 days. These findings emphasize the complex effects of source region, seasonality, and wet removal on varying rBC distributions in the outflow region. We believe that these findings offer valuable observational constraints for improving the physical realism of models.



## 1 Introduction

The pace of recent climate change has intensified, prompting extensive efforts across scientific, political, and societal domains to curb its impacts (IPCC WG1 AR6). However, the nonlinear and interconnected nature of the climate system presents substantial challenges to accurately predicting outcomes and implementing effective mitigation strategies (IPCC WG1 AR6; Steffen et al., 2018). In this regard, short-lived climate forcers (SLCFs), which include species like methane ($CH_4$), ozone ($O_3$), and aerosols, are gaining scientific and political attention in addition to long-lived climate forcers (LLCFs) like $CO_2$. Black Carbon (BC) is a significant SLCF that contributes roughly 0.14 W $m^{-2}$ of radiative forcing. BC is a carbonaceous aerosol that is directly released into the atmosphere from incomplete combustion of biomass, fossil fuels, and biofuels. It absorbs light strongly in the visible and near-infrared spectrum (IPCC WG1 AR6). BC also modifies atmospheric processes indirectly by acting as cloud condensation nuclei (CCN) (Bond et al., 2013; Jacobson, 2014; Kuwata et al., 2008, 2009) and recent studies have reported a linkage between BC and adverse human health effects including cardiovascular and respiratory diseases (Geng et al., 2013; Hvidtfeldt et al., 2019; Kirrane et al., 2019).

Global anthropogenic BC emissions were estimated at 4,741 Gg $year^{-1}$ in 2022, with approximately 49% (2,316 Gg $year^{-1}$) originating from Asia, based on EDGAR v8.1 (Emissions Database for Global Atmospheric Research) inventory (Crippa et al., 2024). Historical trends prior to 2000 show a decline in BC emissions in North America and Europe (Eckhardt et al., 2023), whereas more recent estimates indicate a sustained increase in Asian emissions over the past two decades in ECLIPSE (Evaluating the Climate and Air Quality Impact of Short-Lived Pollutants) inventory (Klimont et al., 2017). While China has remained the largest national contributor in East Asia (~52 % of Asia's emissions in 2015), countries in South Asia and Russia have emerged as increasingly significant contributors to both regional and global BC emissions (Kurokawa and Ohara, 2020; Regional Emission inventory in Asia (REAS) version 3).

Due to their small size typically within the accumulation mode and hydrophobic nature of freshly emitted particles, urban-sourced BC can be transported over long distances, crossing national and even continental boundaries, and impacting remote or polar environments (Deng et al., 2024; Ginot et al., 2014; Lim et al., 2017, 2022). During transport, BC particles undergo aging processes, generally increasing in size via coagulation and becoming more hygroscopic through internal mixing with co-emitted gases and aerosols from combustion sources (Riemer et al., 2010; Utavong et al., 2024; Weingartner et al., 1997; Zuberi et at., 2005). Previous studies have shown that internal mixing and the formation of coatings enhance BC's light absorption by altering refractive properties (Bond et al., 2013; Lack et al., 2009; Liu et al., 2015). Given that both refractive index and the particle's morphology including size, shape, and coating directly determine its light-absorbing ability (Fuller et al., 1999), understanding the chemical and physical nature of BC and its transformation during atmospheric aging is thus essential for characterizing its optical behavior in a regional scale and constraining its representation in models.

Recent aircraft-based observations have revealed diverse vertical and regional characteristics of BC particles. For example, BC particle diameters were found to be smaller near the surface over urban areas (Lamb et al., 2018), whereas in rural regions, BC tended to decrease in size with increasing altitude (Lu et al., 2019). Differences in BC



mass distribution have also reported between the Atlantic and Pacific Oceans (Katich et al., 2018), and substantial
removal of BC, up to 98 %, was observed in Asian summer monsoon outflow (Berberich et al., 2025). These findings
provide valuable insight into the transport efficiency and the lifetime of BC under real-world atmospheric conditions.
Given the complex influence of combustion on Asian air quality, analyzing the distribution of combustion-
derived components can help identify fuel types, combustion efficiency, and removal processes. Carbon monoxide
(CO) shares common sources with BC, making the BC/CO ratio a useful indicator of emission characteristics and
transport efficiency. The BC/CO ratio is typically higher for biomass and diesel emissions, and lower for urban
gasoline combustion, reflecting differences in fuel type and combustion conditions (Bond et al., 2004; Girach et al.,
2014; Zhou et al., 2009). Because BC and CO differ in atmospheric lifetime (about a week and a month, respectively),
the BC/CO ratio is particularly sensitive to BC removal processes. Kanaya et al. (2016) reported that, at an Asian
background site, an accumulated precipitation along trajectory (APT) of $25.5 \pm 6.1$ mm reduced BC transport
efficiency to $1/e$. Similarly, Berberich et al. (2025) showed BC is nearly completely removed from uplifted air in Asian
summer monsoon outflow, based on BC and CO relationship. In addition, the ratio of CO to $CO_2$ has been used as an
indicator of combustion efficiency at the emission source. High-efficiency sources, such as modern power plants,
exhibit $CO/CO_2$ ratios below 0.1 % (Peischl et al., 2010), whereas low-efficiency combustion, such as biomass burning,
results in substantially higher $CO/CO_2$ ratios (Suntharalingam et al., 2004; Wang et al., 2010). According to KORea-
United States-Air Quality (KORUS-AQ) observations, the $CO/CO_2$ ratio in the Korean outflow typically ranged from
0–2 %, whereas the ratio for air masses originating from outside the region and mainland China exhibited a broader
range of 2–4 % (Halliday et al., 2019). Keeping analyzing these tracer ratios is beneficial in monitoring combustion
activities in the region.
Most previous aircraft studies over Asia have focused on observations conducted primarily over inland urban
areas, with limited coverage of marine or downwind regions. Several aircraft campaigns have targeted the Yellow Sea
region of East Asia, including the Aerosol Radiative Forcing in East Asia (A-FORCE; Oshima et al., 2012; March–
April 2009), A-FORCE 2013 winter (Kondo et al., 2016; February–March 2013), and the KORUS-AQ (Lamb et al.,
2018; from May–June 2016). While these campaigns have provided valuable datasets, most airborne measurements
have been limited to short-term, intensive observational periods. This has restricted the ability to comprehensively
assess seasonal variability in the region.
To address this limitation, we present results from a series of aircraft BC measurements conducted over the
Yellow Sea in multiple seasons, from February 2021 to May 2022. BC data were collected aboard the research platform
as part of the Yellow Sea-Air Quality (YES-AQ) campaign. The analysis presented here focuses on vertical and
seasonal variations, combustion characteristics related to air mass origins, and the transport efficiency of BC by
examining its physical properties in conjunction with tracer ratios.



## 2 Methodology

### 2.1 Aircraft Measurements

#### 2.1.1 Overview of Aircraft Measurements

Aircraft measurements were conducted over the Yellow Sea ("YS") between 8 February 2021 and 2 May 2022, using a King Air 350HW (Beechcraft, USA) research aircraft. A total of 23 flights were carried out along a regular flight path spanning latitudes 34.8° N to 37.6° N and longitudes 124.2° E to 127.1° E. The aircraft operated at altitudes ranging from 400 m to 5,000 m to capture the vertical distribution and seasonal variation of BC properties in the region (Fig. 1a). Each flight was labeled as F(flight)+YYMMDD according to the flight date and corresponding flight information is summarized (Table S1). To minimize the influence of aircraft and airport emissions during takeoff and landing, data collected east of 126.5° E were excluded from the analysis.

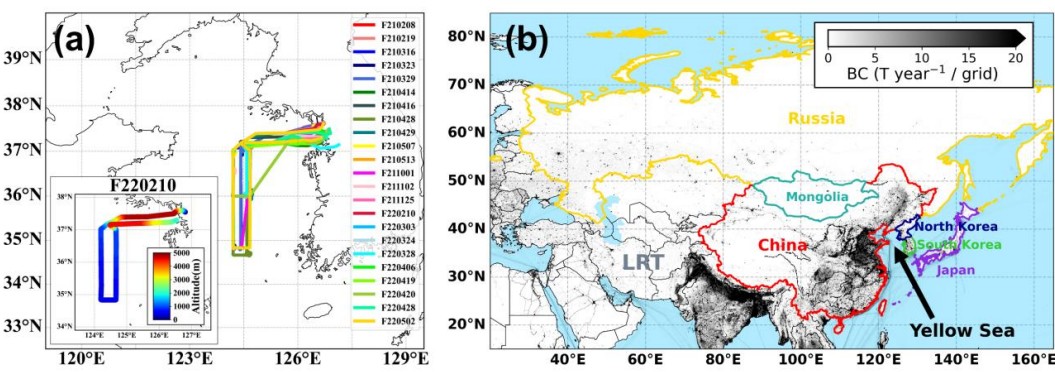

**Figure 1.** (a) Flight paths for all 23 research flights conducted in this study. All flights departed from the location marked with a star. Different colors represent individual flights. The bottom-left panel shows the flight track for F220210, color-coded by altitude as a representative example of the vertical flight profiles. (b) Black carbon emission rates (tons per year / 0.1° × 0.1° grid cell) with regions where air masses influencing the observations during the aircraft mission (South Korea, North Korea, Japan, China, Mongolia, Russia, and LRT in different colors). BC emissions are sourced and averaged from EDGARv8.1 (Crippa et al., 2024) data for 2021 and 2022.

#### 2.1.2 Planetary boundary layer height

Planetary boundary layer height (PBLH) was defined for each aircraft measurement. Each flight was divided into outbound (start to midpoint) and inbound (midpoint to end) segments as it was cycled. This allowed for distinct PBLH estimates to account for diurnal expansion and contraction. Heffter (1980) proposed that a critical inversion can be identified when the vertical gradient of potential temperature ($\Delta\theta/\Delta z$) exceeds 5 K km$^{-1}$. Based on this criterion, Halliday et al. (2019) determined PBLH using $\Delta\theta/\Delta z \geq 6$ K km$^{-1}$ and $\Delta RH$ (RH$_{TOP}$ - RH$_{BASE}$) $\leq$ -5 %. Kang et al. (2021) defined the Entrainment Zone (EZ) for each flight as the 90th percentile segment of $\Delta\theta/\Delta z$ (7–13 K km$^{-1}$). Zhao et al. (2019) set PBLH as the altitude at which $\Delta\theta/\Delta z$ initially reached 5 K km$^{-1}$.



128   In this study, $\Delta\theta/\Delta z$ and $\Delta$RH of each layer were calculated in 50-meter altitude bins. For some flights, larger

129 bin sizes (75 m or 100 m) were used to minimize noise in vertical profiles (Figs. S1 and S2). By this approach, the

130 layers satisfying $\Delta\theta/\Delta z > 5$ K km$^{-1}$ were first selected. Among these candidate layers, the one exhibiting the most

131 negative $\Delta$RH value was designated as the EZ. However, for eight flight cases, rather than choosing the layer with the

132 most negative $\Delta$RH, the layer exhibiting the largest positive $\Delta\theta$ value was selected. We considered an upper limit of

133 1,800 m for base of EZ (=PBLH). Determining PBLH was challenging in some flights mostly due to insufficient data

134 (i.e., F210316 outbound, F210323 inbound, F210329 inbound, F220420 inbound, and F220502 outbound).

## 2.2 Single particle soot photometer

136   Single Particle Soot Photometer (SP2, Droplet Measurement Technologies) uses a laser-induced

137 incandescence technique to measure refractory black carbon (rBC) on a particle-by-particle basis (Bond et al., 2013;

138 Petzold et al., 2013; Schwarz et al., 2010; Stephens et al., 2003). The rBC mass is linearly proportional to the signal

139 from the incandescence channels. Aquadag (Acheson Inc.) was used as the calibration standard for the incandescence

140 channels (Gysel et al., 2011; Moteki and Kondo, 2010), with a differential mobility analyzer to select particle diameters

141 ranging from 80 to 480 nm. Polystyrene Latex spheres (Thermo Scientific) with diameters of 200, 240, 300, and 350

142 nm served as calibration standards for the scattering channels. The SP2 used in this study had a measurement range

143 for rBC of approximately 0.3–130 fg.

144   The mass-equivalent diameter of rBC (rBC$_{MED}$) was calculated from particle mass assuming a void-free

145 density of 1.8 g cm$^{-3}$, and ranged approximately from 70 to 510 nm. (Bond et al., 2013; Moteki and Kondo, 2010).

146 The mass median diameter (MMD) of rBC was determined by fitting a lognormal distribution to the mass size

147 distribution (dM/dlogD$_p$). MMD was calculated as the geometric mean ± geometric standard deviation of 5-minute

148 interval values for each flight, each season, and the entire dataset. In contrast, for specific subsets such as PBL, vertical

149 profiles, Major Regions, and APT conditions, all available data within each category were used to compute a single

150 representative MMD value. The ambient rBC mass concentration (M$_{rBC}$) dataset was converted to standard

151 temperature and pressure (273.15 K, 1,013 hPa). For purely scattering (rBC-free) particles, the SP2 measurable

152 diameter range was approximately 180–470 nm based on calibrations.

153   To determine the mixing state of rBC particles based on the SP2 measurements, either the "Delay time"

154 method or the "Leading Edge Only (LEO)-fit" method is commonly used. For bare (uncoated) rBC particles, the peaks

155 of the incandescence and scattering channel signals appear almost simultaneously. However, when rBC particles are

156 coated, a time difference emerges between the two signal peaks, known as "delay time". This delay time allows for

157 the classification of rBC particles as either "bare or thinly-coated" or "thickly-coated" (Krasowsky et al., 2016; Moteki

158 and Kondo, 2007). In this study, a 1.5 μs delay time threshold was selected from the distribution (Fig. S3) to calculate

159 the number fraction of thickly-coated rBC particles (F$_{thick}$) among total rBC particles measured, as an indicator of rBC

160 mixing state.

161   The LEO fitting method quantifies the coating thickness of individual rBC-containing particles by combining

162 SP2 measurement signals with Mie scattering theory (Gao et al., 2007). In the SP2 laser chamber, the coating of an

163 rBC-containing particle evaporates before the particle reaches the beam center, where scattering is maximized. As a





result, the measured scattering signal is lower than that corresponding to the particle's original (pre-evaporation) size.
The LEO-fit method reconstructs the scattering signal as if the particle retained its initial shell diameter. This
reconstruction is performed by fitting the leading edge of the measured scattering signal, assuming a spherical core-
shell morphology with a refractive index of 2.26 + 1.26i for the rBC core (Moteki et al., 2010) and 1.50 + 0.00i for
the coating (Taylor et al., 2015; Laborde et al., 2012). In this study, the LEO-fit analysis was applied to rBC particles
with diameters ($rBC_{MED}$) between 140 nm and 220 nm, the range in which the method provides the highest accuracy.
The ratio of the reconstructed shell diameter to the measured rBC core diameter ($R_{shell/core}$) was calculated using the
optical shell diameter derived from LEO fitting and the $rBC_{MED}$. All rBC parameters were derived from individual
particle-level data (1 Hz resolution) and were aggregated into 10-second intervals for analysis.

### 2.3 Other measurements

The ambient CO and $CO_2$ concentrations were measured using Cavity Ring Down Spectroscopy (CRDS, G-
2401m, Picarro Inc.). Slopes of rBC versus CO (rBC/CO) and CO versus $CO_2$ ($CO/CO_2$) were obtained and analyzed
along with rBC physical properties after data was averaged at 10-second intervals (Sect. 3.3). In addition, $O_3$ (Thermo
Scientific model 49i), $NO_2$ (Thermo Scientific model 42i-TL), $SO_2$ (Thermo Scientific model 43i-TLE) and aerosol
scattering coefficients at a wavelength of 550 nm (TSI model 3563) were measured simultaneously. Meteorological
parameters, including air temperature, pressure, wind direction, wind speed, and relative humidity, were measured
using the Aircraft Integrated Meteorological Measurement System (AIMMS-20, Aventech Research Inc., Canada).
Aircraft position data (latitude, longitude, altitude) were obtained via a Global Positioning System (GPS, C2626,
Trimble Inc., USA) in the National Marine Electronics Association (NMEA) format.

### 2.4 Airmass backward trajectory analysis and source-region identification

Airmass backward trajectories were analyzed using the Hybrid Single Particle Lagrangian Integrated
Trajectory (HYSPLIT) model provided by the National Oceanic and Atmospheric Administration (Draxler and Hess,
1997, 1998; Stein et al., 2015) for the flight observations. Using meteorological data from the Global Data
Assimilation System (GDAS1, 1° × 1° resolution), airmass trajectories were calculated every 10 seconds for 120 hours,
consistent with the atmospheric lifetime of BC. In addition, the accumulated precipitation along trajectory (APT; in
mm) was calculated by summing the total precipitation over the 72 hours prior to each trajectory endpoint, in order to
investigate the influence of wet removal on rBC properties (Choi et al., 2020b; Kanaya et al., 2016).
To trace the origin of air masses reaching the YS during the aircraft mission, the location of trajectories
endpoints for the past 5 days was assigned as potential source region within administrative area maps from Global
Administrative Areas (http://gadm.org/). The country most frequently crossed by each trajectory below an altitude of
2.5 km was designated as the origin of that air mass. This classification, referred to as "Major Region", was intended
to account for the upward dispersion of surface-emitted pollutants. The Major Region was classified into eight groups:
South Korea, North Korea, Japan, China, Mongolia, Russia, long range transport (LRT) and Ocean (Fig. 1b, Table S3,
and Fig. S4). For each trajectory, the Major Region was determined using the following criteria: 1) Assign the region
crossed most frequently, unless it is Ocean, in which case assign the second most frequently crossed region. 2) If the



most frequent region contributes less than 5 % of endpoints, reassign the Major Region as Ocean to improve
classification reliability. 3) If the trajectory crossed only Ocean, assign the Major Region as Ocean. This correction
was applied to minimize the oceanic bias that may arise from conducting the flights primarily over the sea.
**3 Results and Discussion**
**3.1 Measurement overview of airborne rBC particles**

Throughout the campaign, the YS, located in the mid-latitudes of the Northern Hemisphere, experienced
prevalent westerly winds frequently bringing air masses from the continent. Figure 2 presents the statistical
distribution of observational parameters for each aircraft measurement. The average rBC mass concentration ($M_{rBC}$)
for individual flights ranged from 58.8 to 671.1 ng m$^{-3}$, varying by a factor of 10. The overall campaign mean $\pm$
standard deviation was 210.7 $\pm$ 247.6 ng m$^{-3}$. The lowest flight-mean $M_{rBC}$ (58.8 $\pm$ 104.4 ng m$^{-3}$; F210316) was
comparable to levels measured during aircraft observations over remote regions of continental Europe ($\sim$50 ng m$^{-3}$)
(McMeeking et al., 2010), but clearly higher than the those observed over the northern Greenland Sea (7–18 ng m$^{-3}$;
Ohata et al., 2021). In contrast, the highest flight-mean $M_{rBC}$ (671.1 $\pm$ 492.3 ng m$^{-3}$; F220210) was similar to airborne
observations over the southeastern Indo-Gangetic Plain (700 ng m$^{-3}$; Brooks et al., 2019). This level was approximately
four times higher than the concentrations reported over the Los Angeles Basin (167 $\pm$ 83 ng m$^{-3}$; Metcalf et al., 2012).





**Figure 2**. Variations in Major Region contributions and rBC properties for each flight measurement. (a) Number of data points from the Major Regions, (b) rBC mass concentration ($M_{rBC}$), (c) rBC mass median diameter (MMD), (d) Number fraction of thickly-coated rBC particles ($F_{thick}$), (e) Range of shell-to-rBC core diameter ($R_{shell/core}$), (f) CO concentration, (g) $CO_2$ concentration. In box plots, whiskers extend to 1.5 times the interquartile range, boxes represent the 25th to 75th percentiles, and black dots indicate the mean values for $M_{rBC}$, $F_{thick}$, and $R_{shell/core}$, while for MMD, they represent the geometric mean. Colors indicate specific events; Haze (red), Asian Dust (blue), and combined Asian Dust & Haze (green).



223   Ground-based $M_{rBC}$ levels, particularly in urban areas, have generally been much higher than those observed

224 in this study. In the Seoul megacity, summertime $M_{rBC}$ levels were $480 \pm 290$ ng m$^{-3}$ (Lim et al., 2023), which is

225 roughly twice the mean $M_{rBC}$ observed during this study's aircraft campaign. $M_{rBC}$ measured in Paris, France ($900 \pm$

226 $700$ ng m$^{-3}$), London, UK ($900$–$1,740$ ng m$^{-3}$), Shanghai, China ($3,200$ ng m$^{-3}$), and Xi'an, China (~$9,900$ ng m$^{-3}$)

227 (Gong et al., 2016; Laborde et al., 2013; Liu et al., 2014; Wang et al., 2014) underscore the substantially elevated

228 concentrations observed in highly urbanized regions. Overall, $M_{rBC}$ levels observed over the YS were, on average,

229 comparable to those over European remote regions, although pollution plumes occasionally elevated its levels as

230 similar as those found over the highly polluted Asian regions.

231   The 5-minute interval mass median diameter (MMD) of rBC particles varied dynamically ranging from 70

232 nm to 250 nm. The MMDs of each flight ranged from 158.6 nm to 206.6 nm, with a geometric mean ($\pm$ geometric

233 standard deviation) of $182.3 \pm 1.1$ nm across all flights. Previous studies have shown that observed size distributions

234 of rBC generally reflect both emission sources and further particle physical processes (e.g., coagulation and

235 scavenging) in the atmosphere. As examples from ground-based measurements, in Seoul dominated by urban traffic

236 emissions, the MMD was $127 \pm 11$ nm (Lim et al., 2023), which was smaller than those observed at Jungfraujoch

237 ($220$–$240$ nm; Liu et al., 2010) or in biomass-burning plumes over urban Shanghai (~$200$ nm; Gong et al., 2016). In

238 airborne measurements, MMDs in European remote regions ($180$–$200$ nm) were notably larger than those measured

239 in urban outflow ($170 \pm 10$ nm) (McMeeking et al., 2010). Overall, BC's MMDs tend to be larger in remote regions

240 than in urban areas, and biomass-burning emissions generally produce larger MMDs than traffic-related emissions

241 (Ko et al., 2020; Kompalli et al., 2020, Schwarz et al., 2008). While the MMDs observed over the YS thus reflect

242 characteristics of both urban outflow and remote regions, their wide temporal variability suggests clear influences of

243 emissions from diverse emission sources of multiple countries.

244   The flight-averaged number fraction of thickly-coated rBC particles ($F_{thick}$) varied greatly from 0.31 to 0.81,

245 with a campaign mean of $0.63 \pm 0.16$. In urban regions, $F_{thick}$ values are generally below 0.5, as shown by previous

246 measurements in Houston, USA ($0.09 \pm 0.06$; Schwarz et al., 2008) and Los Angeles ($0.05 \pm 0.02$; Krasowsky et al.,

247 2016), and Xi'an, China (0.48 during polluted periods and 0.38 during clean periods; Wang et al., 2014), indicating

248 that roughly more than 50 % rBC particles are bare from fresh emissions. In the PBL over Beijing, $F_{thick}$ was reported

249 to be approximately 0.5 on average (Zhao et al., 2019), which is slightly lower than mean value observed in this study

250 (0.63). Compared to these previous observations, rBC particles over the YS were more aged on average than those

251 over Beijing, while characteristics of urban outflow were revealed. It is noteworthy that two mixing state parameters

252 ($F_{thick}$ and $R_{shell/core}$) showed similar patterns (Fig. 2). Flight-averaged $R_{shell/core}$ values ranged from 1.23 to 1.55, with a

253 campaign mean of $1.35 \pm 0.14$. As expected, $R_{shell/core}$ values were higher than those observed at urban ground sites,

254 such as $1.25 \pm 0.07$ in summertime Seoul (Lim et al., 2023) and 1.2 in Beijing from May to June (Liu et al., 2020).

255 However, the degree of BC internal mixing was clearly lower than observed over the Southeast Asia Sea near heavily

256 polluted regions ($R_{shell/core} > 2$, Kompalli et al., 2021).

257   Thus, both size distributions (MMD) and mixing state ($F_{thick}$ and $R_{shell/core}$) of rBC particles observed in this

258 study clearly indicate their considerable dependence on the origins and further chemical/physical processes of the air

259 masses during transport to this remote environment.





### 3.2. Seasonally-varying vertical distributions of rBC properties

#### 3.2.1 Seasonal variability

A total of 23 flights were grouped into five categories by seasons: 2021 Winter (February–March 2021, n=5 flights), 2021 Spring (April–May 2021, n=6 flights), 2021 Autumn (October–November 2021, n=3 flights), 2022 Winter (February–March 2022, n=4 flights), and 2022 Spring (April–May 2022, n=5 flights). Seasonally, $M_{rBC}$ was the highest in winter. In 2022, the winter mean reached $394.3 \pm 407.7$ ng m$^{-3}$, followed by autumn ($278.9 \pm 226.0$ ng m$^{-3}$), and spring ($239.0 \pm 200.0$ ng m$^{-3}$). In contrast, in 2021, winter and spring exhibited similar mean values, though winter showed greater variability (Table 1). This seasonal pattern is consistent with previous studies in East Asia (Kanaya et al., 2020; Liu et al., 2019, 2022; Lim et al., 2012; Zhao et al., 2013).

The elevation of cold-season $M_{rBC}$ levels reflect the combined effects of increased combustion emissions (e.g., heating) and the synoptic-scale meteorology (predominant westerly winds; Fig. S5), which facilitate the efficient transport of rBC from continental source regions to the YS (Gandham et al., 2022; Zhang et al., 1997). Supporting this, anthropogenic BC emissions (from biofuel and fossil fuel sources) based on the EDGAR v8.1 inventory were highest in 2022 Winter across South Korea and its neighboring countries (North Korea, Japan, China, Mongolia, and Russia; Table S4). The remarkable enhancement of $M_{rBC}$ for 2022 Winter can thus be primarily attributed to intensified regional combustion activities. Furthermore, seasonally-averaged aerosol optical depths (550 nm, DT/DB combined) retrieved along 10-second air mass back trajectories below 2,500 m altitude (Fig. S6) exhibited a consistent seasonal pattern with $M_{rBC}$, reinforcing the link between regional aerosol loading and combustion activity.





**Table 1.** Seasonal $M_{rBC}$, MMD, $F_{thick}$, and $R_{shell/core}$ (mean ± std). The "FT" in this study is defined as altitudes reaching up to
approximately 5 km.

| Season | | $M_{rBC}$ (ng m$^{-3}$) | MMD (nm) | $F_{thick}$ | $R_{shell/core}$ |
|---|---|---|---|---|---|
| All (PBLH$^*$: 880±371 m) | all | 210.7±247.6 | 182.3±1.1 | 0.63±0.16 | 1.35±0.14 |
| | FT | 99.8±134.1 | 186.2 | 0.60±0.18 | 1.32±0.13 |
| | PBL | 329.0±264.7 | 193.2 | 0.66±0.14 | 1.39±0.12 |
| 2021 Winter (PBLH: 1,011±417 m) | all | 112.9±112.3 | 182.2±1.1 | 0.48±0.20 | 1.28±0.14 |
| | FT | 36.5±36.7 | 177.8 | 0.45±0.22 | 1.26±0.15 |
| | PBL | 215.3±88.9 | 188 | 0.53±0.17 | 1.31±0.10 |
| 2021 Spring (PBLH: 685±359 m) | all | 110.8±82.0 | 172.1±1.1 | 0.58±0.14 | 1.34±0.08 |
| | FT | 97.4±81.5 | 173.3 | 0.57±0.14 | 1.33±0.08 |
| | PBL | 128.3±59.1 | 183.6 | 0.62±0.14 | 1.38±0.06 |
| 2021 Autumn (PBLH: 1,144±258 m) | all | 278.9±226.0 | 161.9±1.1 | 0.70±0.09 | 1.46±0.13 |
| | FT | 58.0±85.5 | 161.6 | 0.68±0.14 | 1.37±0.17 |
| | PBL | 432.7±162.9 | 174.6 | 0.71±0.04 | 1.51±0.05 |
| 2022 Winter (PBLH: 816±316 m) | all | 394.3±407.7 | 201.4±1.1 | 0.71±0.12 | 1.40±0.17 |
| | FT | 142.1±213.7 | 204.5 | 0.68±0.13 | 1.32±0.16 |
| | PBL | 732.4±338.4 | 207.1 | 0.76±0.11 | 1.50±0.14 |
| 2022 Spring (PBLH: 873±464 m) | all | 239.0±200.0 | 194.4±1.1 | 0.70±0.07 | 1.33±0.11 |
| | FT | 169.2±166.0 | 193.6 | 0.72±0.08 | 1.33±0.12 |
| | PBL | 250.0±176.9 | 197.8 | 0.69±0.07 | 1.31±0.09 |

$^*$ See Sect. 2.1.2. for the PBLH method.

MMD exhibited seasonal variations similar to $M_{rBC}$, with the largest rBC size in winter (190.9 ± 1.1 nm),
followed by spring (181.3 ± 1.1 nm) and autumn (161.9 ± 1.1 nm). In Asia, the size distribution of rBC particles
typically varies seasonally, with larger rBC in cold season than in warm season (Kompalli et al., 2020; Wu et al., 2021;
Yang et al., 2019), reflecting an enhanced influence of both biomass burning emissions and long-range transport effects
during cold season. The MMD difference was about 20 nm between winter and autumn in this study, comparable to
the winter (210–220 nm) to late spring (190–215 nm) difference observed over Beijing (Zhao et al., 2019). This
suggests that the seasonal mean MMD difference can be about 20 nm in East Asia, likely stemming from varying
major emission sources.
Notably, the particularly low levels of internal mixing and $M_{rBC}$ observed in 2021 Winter were likely
associated with reduced anthropogenic activity during the COVID-19 pandemic. In contrast to the relatively consistent
seasonal patterns of $M_{rBC}$ and MMD, the mixing state of rBC particles ($F_{thick}$ and $R_{shell/core}$) was more complex



seasonally and displayed pronounced interannual differences. Specifically, in Table 1, $F_{thick}$ was highest in 2022 Winter
$(0.71 \pm 0.12)$, followed by 2022 Spring $(0.70 \pm 0.07)$, 2021 Autumn $(0.70 \pm 0.09)$, 2021 Spring $(0.58 \pm 0.14)$, and
2021 Winter $(0.48 \pm 0.20)$. In contrast, $R_{shell/core}$ was highest in 2021 Autumn $(1.46 \pm 0.13)$, followed by 2022 Winter
$(1.40 \pm 0.17)$, 2021 Spring $(1.34 \pm 0.08)$, 2022 Spring $(1.33 \pm 0.11)$, and 2021 Winter $(1.28 \pm 0.14)$. These patterns
suggest that the degree of rBC mixing state is influenced not only by seasonal emission strength but significantly also
by meteorological factors operating at both synoptic- and microscale levels. Overall, while the enhanced values of
$M_{rBC}$, MMD, and $F_{thick}$ in winter were commonly found, inter-annual variability merits further discussion in Sect. 3.3.
Figure 3 shows seasonal relationships between planetary boundary layer height (PBLH) and rBC mass
concentration ($M_{rBC}$) within the PBL. Overall, $M_{rBC}$ within the PBL decreased with increasing PBLH, with a more
pronounced pattern in winter (-23.4 ng m$^{-3}$/$\Delta$100 m and -97.6 ng m$^{-3}$/$\Delta$100 m for 2021 and 2022) than spring (-12.8
ng m$^{-3}$/$\Delta$100 m and -26.4 ng m$^{-3}$/$\Delta$100 m for 2021 and 2022). This finding demonstrates the stronger sensitivity of
$M_{rBC}$ distribution on PBL development.

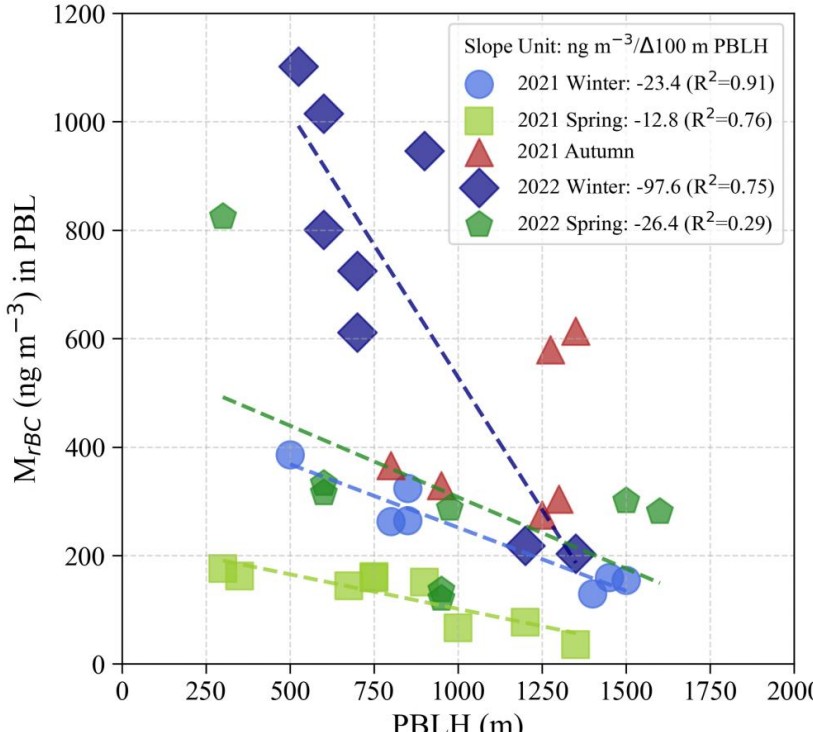

**Figure 3.** Seasonal relationship between planetary boundary layer height (PBLH) and $M_{rBC}$ within the PBL. Each point represents
$M_{rBC}$ averaged within the PBL for either the outbound or inbound segment. Seasonal linear regression lines are shown
for Winter and Spring, with slope values expressed in ng m$^{-3}$ per 100 m of PBLH.



### 3.2.2 Vertical distribution


Figure 4 illustrates seasonally-varying vertical profiles of rBC properties. $M_{rBC}$ exhibited vertical variability
spanning up to two orders of magnitude between the near sea surface and 5 km, with an exponential fit of $x(y) = 40.5$
$+ 571.6 \times exp(-0.87y(km))$. Throughout the experiment period, the average $M_{rBC}$ in the PBL ($M_{rBC\_PBL}$; $329.0 \pm 264.7$
ng m$^{-3}$) was more than three times greater than that in the lower free troposphere ($M_{rBC\_FT}$; $99.8 \pm 134.1$ ng m$^{-3}$,
hereafter referred to as "FT"). Seasonally, the $M_{rBC\_PBL}/M_{rBC\_FT}$ ratios tended to increase from spring (~1.4) to winter
(~5.6). Stronger convective uplift effect may partly explain the smaller ratio in spring.
It is noteworthy that both size and internal mixing of rBC decreased with altitude (-4.64 nm km$^{-1}$ for MMD,
-0.035 km$^{-1}$ for $F_{thick}$, and -0.027 km$^{-1}$ for $R_{shell/core}$, respectively). But seasonality was different between two physical
properties. The decreasing trends were clearer in spring for MMD (-5.63 nm km$^{-1}$) but in winter for $F_{thick}$ and $R_{shell/core}$
(-0.045 km$^{-1}$ and -0.034 km$^{-1}$). Lim et al. (2017) reported that rBC particles preserved in high-altitude (~5.1 km) alpine
ice cores exhibited larger MMDs (~200–300 nm) than found on the ground, suggesting that larger rBC particles in the
atmosphere are preferentially removed by precipitation and subsequently deposited in ice cores. In a remote marine
region far from BC emission sources, markedly thinner coatings of rBC particles were observed, attributing this to
preferential scavenging, photolysis, limited precursor supply, and dilution of continental outflow (Kompalli et al.,
2021). In line with these studies, our findings reflect that both size and mixing state of rBC particles are strongly
sensitive to wet scavenging (cloud and precipitation) at high altitudes (above ~3 km), where rBC particles undergo
long-range transport and have a greater likelihood of being scavenged upon transport. It is noteworthy that cloud
scavenged larger and mixed rBC particles might have contributed to CCN over YS or potentially further into the
Pacific. Yet, the detailed processes governing the observed vertical profiles of rBC physical properties remain unclear.
Also, enhanced photochemical reaction at upper altitude may explain less steeper decreasing pattern of
particle internal mixing in spring. As an interesting seasonal feature, 2022 Spring was the only season in which both
$M_{rBC}$ and $F_{thick}$ were relatively higher in the FT than in the PBL. Given the high temperature ($5.1 \pm 10.0$ °C) and $O_3$
concentrations ($58.9 \pm 9.8$ ppbv) in the FT, these conditions likely facilitated enhanced photochemical reactions aloft,
promoting rBC coating formation and leading to higher degree of rBC mixing state.





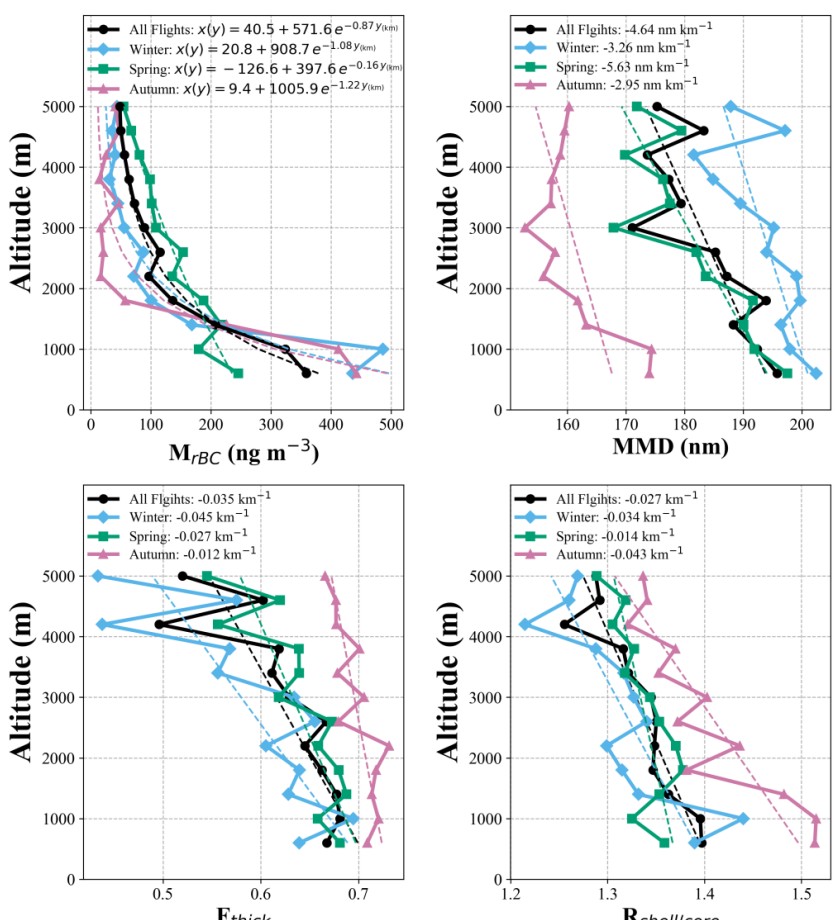

**Figure 4.** Seasonally varying vertical profiles of rBC particles ($M_{rBC}$, MMD, $F_{thick}$, and $R_{shell/core}$). Dashed lines show an exponential fit for $M_{rBC}$ and linear regression fits for the others.

### 3.3 Distinguishing combustion source-region characteristics

The physical and chemical characteristics of combustion plumes reaching the YS were examined by analyzing rBC properties, rBC/CO ratios, and $CO/CO_2$ ratios in relation to air mass origin (Table 2; Fig. 5). Among the various source regions, those originating from South Korea and Japan, captured in 2021 Spring only, exhibited the smallest MMDs (163.4 nm and 168.4 nm, respectively). These MMDs are consistent with those reported for urban outflow (Cho et al., 2021; Lamb et al., 2018; McMeeking et al., 2010; Schwarz et al., 2008), suggesting a dominant influence of urban fossil-combustion emissions. In addition to smaller rBC particle sizes, South Korea- and Japan-sourced air masses also exhibited relatively low rBC/CO slopes, recorded at $0.67 \pm 0.02$ ng $m^{-3}$ $ppbv^{-1}$ and $1.11 \pm 0.03$ ng $m^{-3}$ $ppbv^{-1}$, respectively. The rBC/CO ratio primarily reflects differences in BC/CO emission ratios across





combustion sources, while also capturing the effects of removal processes during atmospheric transport (Kanaya et
al., 2016; Oshima et al., 2012). The particularly low rBC/CO values for these air masses likely indicate a greater
contribution from gasoline combustion, which emits less rBC relative to CO compared to diesel combustion.

**Table 2.** Summary of rBC properties ($M_{rBC}$, MMD, $F_{thick}$, and $R_{shell/core}$) and APT for each Major Region.

| | $M_{rBC}$ (ng m$^{-3}$) | MMD (nm) | $F_{thick}$ | $R_{shell/core}$ | APT (mm) |
|---|---|---|---|---|---|
| South Korea[a] | 142.1±106.0 | 163.4 | 0.65±0.04 | 1.39±0.05 | 1.77±1.51 |
| North Korea[b] | 87.4±70.6 | 201.3 | 0.71±0.10 | 1.31±0.08 | 1.03±0.79 |
| Japan[c] | 102.6±75.0 | 168.4 | 0.68±0.04 | 1.40±0.05 | 0.56±0.83 |
| China | 351.4±287.7 | 194.3 | 0.67±0.13 | 1.40±0.13 | 0.50±1.19 |
| Mongolia | 260.1±218.9 | 185.3 | 0.63±0.14 | 1.38±0.12 | 0.95±1.28 |
| Russia | 150.4±106.7 | 196.9 | 0.60±0.15 | 1.35±0.10 | 2.11±3.54 |
| LRT | 78.5±122.0 | 181.5 | 0.48±0.21 | 1.26±0.14 | 2.89±4.51 |
| Ocean | 102.6±106.5 | 187.7 | 0.68±0.11 | 1.33±0.12 | 1.30±2.60 |

[a, c] 2021 Spring data only.
[b] Mostly 2022 Spring data (89%)



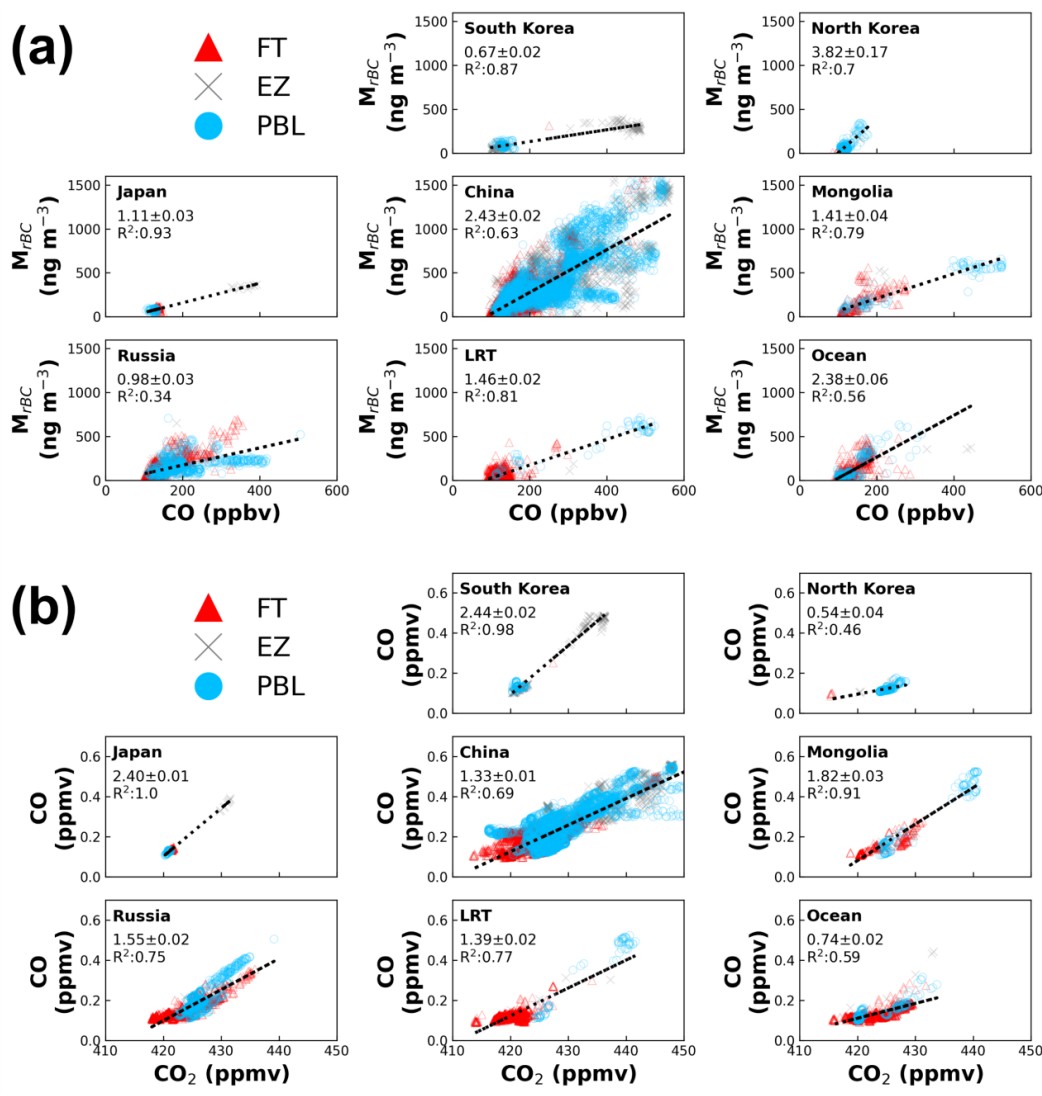

**Figure 5.** Linear relationships of (a) rBC mass concentration ($M_{rBC}$) versus CO and (b) CO versus $CO_2$ in each Major Region, calculated from 10-second averaged data. Sampling layers are denoted by free troposphere (FT) in red triangles, entrainment zone (EZ) in gray crosses, and planetary boundary layer (PBL) in blue circles. The black dashed lines indicate the best-fit linear regressions to all altitude data in each region; the annotated slope and coefficient of determination ($R^2$) quantify the strength and magnitude of each relationship.





Air masses originating from 'North Korea', observed exclusively during spring 2022, exhibited the highest
rBC/CO slope ($3.82 \pm 0.17$ ng m$^{-3}$ ppbv$^{-1}$, $R^2 = 0.70$). It was closely aligned with the springtime KORUS-AQ campaign
value ($3.2 \pm 0.2$ ng m$^{-3}$ ppbv$^{-1}$, $R^2 = 0.83$; Lamb et al., 2018) for North Korean outflow. In both studies, these slopes
were the highest among all regions. Interestingly, this North Korea-sourced air showed the largest MMD (201.3 nm)
and $F_{thick}$ ($0.71 \pm 0.10$), whereas the mean $M_{rBC}$ ($87.4 \pm 70.6$ ng m$^{-3}$) was the second lowest following 'LRT' ($78.5 \pm$
122.0 ng m$^{-3}$). Given the short transport time to the YS ($24 \pm 26$ hours) due to North Korea's geographic proximity,
the largest MMD and rBC/CO slope are more likely attributable to combustion activities associated with biomass and
low-graded coal rather than particle coagulation effect during transport. The mean coating thickness ($30.1 \pm 6.5$ nm)
was the second thinnest after that in LRT air masses ($26.0 \pm 11.5$ nm), suggesting limited atmospheric processing.
Despite relatively high $F_{thick}$, the $R_{shell/core}$ ratio remained moderate ($1.31 \pm 0.08$), likely reflecting the short atmospheric
residence time insufficient for substantial coating development.
China-sourced air exhibited distinct features in rBC properties and tracer characteristics. The rBC/CO slope
was $2.43 \pm 0.02$ ng m$^{-3}$ ppbv$^{-1}$ ($R^2 = 0.63$), in strong agreement with values reported during the KORUS-AQ campaign
($2.2 \pm 0.0$ ng m$^{-3}$ ppbv$^{-1}$, $R^2 = 0.82$; Lamb et al., 2018). Correspondingly, these air masses exhibited elevated levels of
MMD (194.3 nm), $F_{thick}$ ($0.67 \pm 0.13$), $R_{shell/core}$ ($1.40 \pm 0.13$) as well as $M_{rBC}$ ($351.4 \pm 287.7$ ng m$^{-3}$). Regionally, air
from eastern China exhibited the highest $M_{rBC}$ ($442.5 \pm 359.2$ ng m$^{-3}$), MMD (199.3 nm), and $R_{shell/core}$ ($1.43 \pm 0.14$),
while the lowest $M_{rBC}$ ($115.8 \pm 102.9$ ng m$^{-3}$) and MMD (181.1 nm) were found in air masses from less populated
northwestern China. The $CO/CO_2$ slope for 'China' air was substantially higher (3.33 %) than 'Korean Peninsula'
(0.95 %) during the KORUS-AQ (Halliday et al., 2016). However, in this study, the $CO/CO_2$ slope for China air was
$1.33 \pm 0.01$ %, which were similar to those in LRT ($1.39 \pm 0.02$ %), Russia ($1.55 \pm 0.02$ %), Mongolia ($1.82 \pm 0.03$ %).
This relatively lower $CO/CO_2$ slope in this study may reflect rapidly decreasing CO emissions in China (Yan et al.,
2025; Zhao et al., 2024).
Air masses originating from Mongolia, Russia, and LRT consistently delivered larger rBC particles to the
YS. Despite these air masses often experiencing the large amount of precipitation along their trajectories (APT in
Table 2), the persistence of large rBC particle sizes—indicated by sustained high MMD values—suggests that the
dominant sources were biomass and coal combustion, which typically emit larger rBC particles. Given that rBC
particles within the accumulation mode are generally vulnerable to removal by precipitation (Lim et al., 2017), this
observation highlights the robustness of source signatures even after long-range transport.
Both Mongolia- and LRT-sourced air masses were characterized by similar rBC/CO ($1.41 \pm 0.04$ ng m$^{-3}$ ppbv$^{-}$
$^{1}$ and $1.46 \pm 0.02$ ng m$^{-3}$ ppbv$^{-1}$, respectively) and $CO/CO_2$ ($1.82 \pm 0.03$ % and $1.39 \pm 0.02$ %, respectively) slopes.
Compared with China-sourced air masses, the lower rBC/CO slopes suggest that these air masses had undergone more
significant atmospheric aging during transport, while the relatively higher $CO/CO_2$ slopes imply a stronger influence
of incomplete combustion. Further supporting the aging interpretation, the BC/CO emission ratio for Mongolia is
reported as 14.9 ng m$^{-3}$ ppbv$^{-1}$ in EDGAR v8.1 (Table S5; Crippa et al., 2024), indicative of strong fresh biomass
burning emissions at the source. The substantially lower rBC/CO slope measured in this study ($1.41 \pm 0.04$ ng m$^{-3}$
ppbv$^{-1}$) therefore points to considerable loss of rBC during transport. Russian air masses exhibited clear vertical
contrasts. When reaching the PBL over the YS, they exhibited lower rBC/CO slopes and higher $CO/CO_2$ slopes,



whereas the opposite pattern was observed when they arrived at FT. This vertical difference appears to be associated
with elevated CO concentrations in the PBL, potentially influenced by enhanced emissions along the Russian transport
pathway.

**3.4 Wet removal of BC and transport efficiency**

The BC/CO emission ratios reported for source regions in EDGAR v8.1 (i.e., BC emission divided by CO
emission in inventories) were significantly higher than the rBC/CO slopes observed over the YS (Table S5). While the
inventory-based tracer ratio reflects source-specific emission characteristics, the observed slope represents the
transported air mass. This discrepancy primarily indicates preferential wet removal of rBC relative to CO during
transport, assuming similar travel times from the emission source regions.
To evaluate how precipitation influences rBC characteristics remaining in the atmosphere, Table 3 presents
rBC properties and the rBC/CO slope across different APT bins. All rBC-related parameters generally declined with
increasing APT. Most notably, $M_{rBC}$ decreased by more than 50 % at APT ≥ 1 mm relative to dry conditions (APT =
0), accompanied by a substantial reduction in the rBC/CO slope. Although $F_{thick}$ declined by over 10 % at APT ≥ 1
mm relative to dry condition (APT = 0), $R_{shell/core}$ exhibited only a minor reduction (~3 %). This implies that water-
soluble coating materials are not linearly scavenged by the amount of precipitation, but more complex mechanisms
likely govern the wet removal efficiency such as variations in coating composition, size, and the effect of rBC core
morphology.
MMD initially decreased up to APT ≥ 1 mm but rose again to 199.7 nm at APT ≥ 10 mm. Notably, air masses
with APT ≥ 10 mm predominantly originated from Russia, LRT, and Mongolia, and retained larger rBC particles
despite high precipitation. Again, even in the presence of heavy precipitation, the larger rBC size indicates a substantial
influence from combustion sources that can emit larger particles, such as biomass and coal burning.

**Table 3.** Summary of rBC-related properties ($M_{rBC}$, MMD, $F_{thick}$, $R_{shell/core}$, and rBC/CO slope) across different APT bins.

| | All data | APT = 0 | APT ≥ 0.1 mm | APT ≥ 1 mm | APT ≥ 10 mm |
|---|---|---|---|---|---|
| $M_{rBC}$ (ng m$^{-3}$) | 210.7±247.6 | 266.6±286.1 | 161.5±195.2 | 114.8±130.4 | 46.8±68.2 |
| MMD (nm) | 192.4 | 193.9 | 189.8 | 187.4 | 199.7 |
| $F_{thick}$ | 0.63±0.16 | 0.65±0.17 | 0.61±0.16 | 0.58±0.17 | 0.55±0.22 |
| $R_{shell/core}$ | 1.35±0.14 | 1.37±0.14 | 1.34±0.13 | 1.32±0.13 | 1.33±0.21 |
| rBC/CO slope (ng m$^{-3}$ ppbv$^{-1}$) | 2.25 | 2.40 | 1.96 | 1.43 | 1.15 |

To quantify the wet removal efficiency of rBC, its transport efficiency (TE) was estimated based on the
rBC/CO slope under different APT conditions, following the approach of Matsui et al. (2011) and Oshima et al. (2012).
$$Transport\ Efficiency\ (TE) = \frac{[rBC/CO]_{APT>0}}{[rBC/CO]_{APT=0}} ,\qquad(1)$$





By definition, TE is set to 1 at APT = 0. APT was divided into 10 bins, and for each bin, the rBC/CO slope
was calculated and normalized by the slope at APT = 0. $[rBC/CO]_{APT=0}$ was determined to be 2.40 ng m$^{-3}$ ppbv$^{-1}$.
Figure 6 shows the TE as a function of the APT. The blue dashed lines indicate TE values of 1/e and 0.5, and they
were fitted with a stretched exponential decay (SED) function, yielding TE = exp(-0.246 × APT$^{0.478}$) with a $R^2$ of 0.91.
From the fitted curve, the BC TE was found to reduce to half (TE$_{0.5}$) at APT = 8.8 mm and reached 1/e (TE$_{1/e}$) at APT
= 18.9 mm. Assuming an annual precipitation of 1253.7 mm for Seosan (a coastal site near the YS), it was estimated
that rBC would decrease to half and to 1/e over 2.6 days (61.3 hours) and 5.5 days (131.9 hours), respectively. These
findings are broadly consistent with earlier findings. Oshima et al. (2012) reported TE$_{0.5}$ at APT ~10 mm in the East
China Sea, and Kanaya et al. (2016) found TE$_{0.5}$ and TE$_{1/e}$ at 15.0 ± 3.2 and 25.5 ± 6.1 mm, respectively, in Fukue
Island, Japan. Compared to these results, the slightly lower TE$_{0.5}$ and TE$_{1/e}$ thresholds observed in this study suggest
a relatively faster wet removal of rBC over the YS region. This likely reflects differences in air mass characteristics,
precipitation intensity, or physico-chemical BC dynamics. In particular, larger particles and those coated with highly
water-soluble materials are known to be more readily scavenged, possibly contributing to the enhanced removal
efficiency observed here.

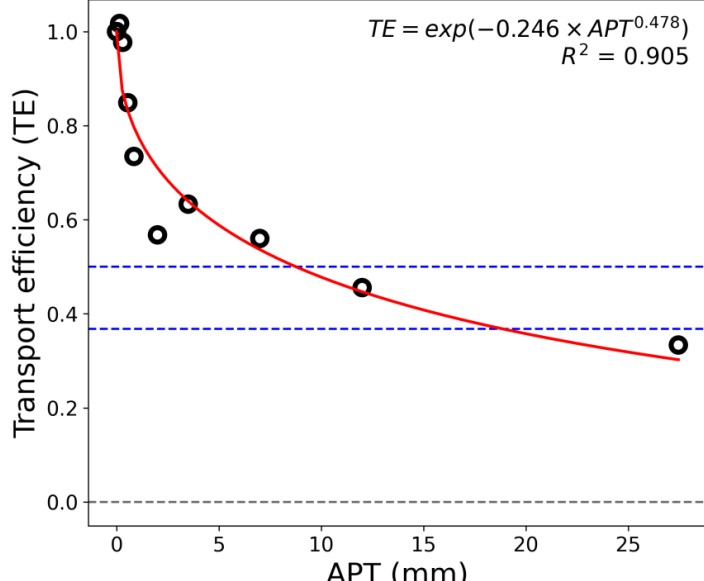


**Figure 6.** Transport efficiency as a function of APT observed over the YS. The blue dashed lines indicate TE values of 1/e and
0.5, respectively.



## 4 Conclusion


To investigate the physical properties of rBC and associated tracer characteristics in continental outflow, a
total of 23 research flights were conducted over the YS between 8 February 2021 and 2 May 2022. Individual flight-
mean $M_{rBC}$ ranged from 58.8 to 671.1 ng m$^{-3}$, varying by more than an order of magnitude, with MMD of 158.6–206.6
nm, $F_{thick}$ of 0.31–0.81, and $R_{shell/core}$ of 1.23–1.55. Notably low levels of $M_{rBC}$ and internal mixing in 2021 Winter
likely resulted from reduced anthropogenic combustion activities during the COVID-19 pandemic.

Seasonally, $M_{rBC}$ and MMD exhibited similar trends, and vertically, $M_{rBC}$ varied by up to two orders of
magnitude between the sea surface and 5 km asl., highlighting the strong stratification in BC mass loading. Although
Lamb et al. (2018) reported smaller surface MMDs near urban areas, such reductions were not observed in this study,
likely due to the exclusion of urban influences in the over-sea dataset. More importantly, the vertical patterns of MMD
and mixing state parameters in our study suggest a stronger influence from wet scavenging processes during particle
transport, particularly at altitudes above ~3 km.

Air masses from South Korea and Japan exhibited low MMD and rBC/CO slopes, indicating a predominant
influence from vehicle emissions. In contrast, China-sourced air masses showed elevated levels of $M_{rBC}$, MMD, $F_{thick}$,
and $R_{shell/core}$, suggesting additional contributions from biomass and coal combustion. The rBC/CO slope of China
($2.43 \pm 0.02$ ng m$^{-3}$ ppbv$^{-1}$) closely matched the value observed during the KORUS-AQ campaign ($2.2 \pm 0.0$ ng m$^{-3}$
ppbv$^{-1}$; lamb et al., 2018), while the lower $CO/CO_2$ ratio likely reflects the continued decline in CO emissions in China.
As increasing precipitation, rBC-related parameters generally decreased, with $M_{rBC}$ reduced to one-fifth at ATP $\geq 10$
mm. The relatively faster wet removal of rBC over the YS, which is estimated to reduce to 1/e within 5.5 days,
highlights the influence of air mass characteristics, precipitation intensity, and particle properties on BC scavenging
efficiency. Such estimates of BC lifetime provide essential constraints for improving wet deposition schemes in
climate models.

In this study, our findings demonstrate rBC physical properties and tracer ratios are effective tools for
distinguishing and monitoring major combustion activities from multiple countries as they retain their original
information. At the same time, rBC concentrations and physical properties exhibited clear vertical and seasonal
patterns, while multifaceted mixing state points to greater diversity in particle-level properties and complexity of
atmospheric processing. To better constrain the atmospheric fate and climate effects of rBC, future research should
focus on resolving the chemical and morphological properties of rBC particles and related atmospheric processes.



**Code/Date availability**

The data of this paper can be obtained from https://doi.org/10.5281/zenodo.15951968.

**Author contributions**

NY and SL designed the methodology, performed the validation, and wrote the original draft. NY also carried out data curation, formal analysis, visualization, and prepared the figures. SL supervised the project, secured funding, managed project administration, and provided resources. HJY, SO, YC, SRL, and SK contributed to manuscript review and editing. Additionally, HJY, SO, SRL, and SK provided resources and contributed to the investigation, while YC supported the methodology and formal analysis. HJY and SK also contributed to funding acquisition. All authors participated in interpreting the results and approved the final manuscript.

**Competing interests**

The authors declare that none of the authors has any competing interest.

**Acknowledgements**

We acknowledge the NOAA Air Resources Laboratory (ARL) for providing the HYSPLIT transport and dispersion model. We also acknowledge the use of black carbon emission inventory data from the Emissions Database for Global Atmospheric Research (EDGAR). In addition, we acknowledge the use of aerosol optical depth (AOD) data from NASA's MODIS (Moderate Resolution Imaging Spectroradiometer) instrument.

**Financial support**

This work was supported by the Korea Meteorological Administration Research and Development Program "Development of Asian dust and haze monitoring and prediction technology (KMA 2018-00521)", the National Research Foundation of Korea (NRF) from the Ministry of Science and ICT (NRF-2021R1C1C2011543 & RS-2023-00218203), and Korea Ministry of Environment (Korea MOE) as Waste to energy recycling Human resource development Project.



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
