# Peer review of "Impacts of Source Regions and Atmospheric Transport on"

_EGUsphere, 2025_

## Author Comment (AC1)

Response to Reviewer Comments

We thank the reviewer for the constructive comments and suggestions, which helped us improve the clarity and quality of the manuscript. In this response letter, we provide point-by-point responses. Our replies to the reviewers' comments are presented **in blue font**, while the corresponding revisions in the manuscript are indicated **in blue italics with yellow highlights, together with the updated line numbers**.

**General comments**

This study presents a unique dataset of aircraft measurements of BC and its microphysical properties over the Yellow Sea, consistently conducted throughout several years. The results will be useful for constraining the assumptions and parameterizations of size-distribution and mixing state of BC in aerosol-climate models and in aerosol remote sensing. I can recommend publication after incorporating the following comments.

**Response:** We sincerely thank the reviewer for the positive assessment and constructive feedback. The manuscript has been carefully and thoroughly revised in accordance with these comments.

**Major comments**

**Comment:**
In Section 1 or Section 3, the authors should add references of observational studies on the relationship between the microphysical properties and wet removal efficiency of BC in the same region, for example, https://doi.org/10.1029/2012GL052034. It should help the interpretation of the decrease of rBC MMD with APT or altitude observed in this study.

**Response:**

We appreciate this valuable suggestion. There have been several insightful studies

which deal with BC's properties in relation to APT and transport efficiency (TE).

Moteki et al. (2012) found, from aircraft observations over the Yellow and East China Seas, that as BC TE decreases, larger-mass BC particles are preferentially removed, resulting in a smaller count median diameter (CMD) during uplift from the planetary boundary layer to the free troposphere. An analysis of the relationship between BC TE and APT revealed a significant negative correlation ($R^2$ = 0.88) at altitudes above 2 km, and TE decreased with increasing APT as altitude increased (Oshima et al., 2012). In line with these studies, seasonal differences in precipitation and uplift strongly influence BC concentrations, and TE decreases as APT increases, indicating that the seasonal variability of BC is largely controlled by the seasonality of wet deposition (Kondo et al., 2016).

In the revised manuscript, we have incorporated the aforementioned references and expanded the discussion on the relationship between BC microphysical properties and wet removal efficiency based on these studies.

L 65–76:

"*Recent aircraft-based observations have revealed diverse vertical and regional characteristics of BC particles. For example, BC particle diameters were found to be smaller near the surface over urban areas (Lamb et al., 2018), whereas in rural regions, BC tended to decrease in size with increasing altitude (Lu et al., 2019). Regional differences in BC mass concentration have also reported between the Atlantic and Pacific Oceans (Katich et al., 2018), and substantial removal of up to 98 % of BC was observed in Asian summer monsoon outflow (Berberich et al., 2025). These findings provide valuable insight into combined effects of emission characteristics, meteorology, and removal processes under real-world atmospheric conditions. More specifically, aircraft observations over the Yellow and East China Seas revealed that BC transport efficiency (TE) declines during uplift from the planetary boundary layer (PBL) to the free troposphere (FT), preferentially removing large-mass BC particles and thus reducing the count median diameter (CMD) (Moteki et al. (2012). At altitudes above 2 km, TE showed a strong negative correlation with accumulated precipitation along air mass trajectory (APT), with $R^2$ = 0.88 (Oshima et al., 2012). Building on this, Kondo et al. (2016) demonstrated that*

*seasonal variations in precipitation and uplift patterns strongly modulate BC concentrations."*

L 369–370:

*"Seasonal differences in precipitation and uplift significantly influence BC concentrations (Kondo et al., 2016) and physical properties (Moteki et al., 2012)."*

**Individual comments**

**L17:** refractive BC → refractory BC

It has been corrected.

**L18:** height-dependency → height-dependence

It has been corrected.

**L24:** reduction → decrease

It has been corrected.

**L25:** These findings emphasize → These observations reflect

It has been corrected.

**L26:** You can remove "We believe that"

It has been removed and revised as follows.

L25–26:

*"These findings provide valuable observational constraints for improving model representations of the size distribution and mixing state of ambient BC particles in the outflow regions."*

**L27:** "the physical realism of models": Please be more specific.

It has been revised to the following sentence.

L 25–26:

"*These findings provide valuable observational constraints for improving model representations of the size distribution and mixing state of ambient BC particles in the outflow regions.*"

**L34:** "IPCC WG1 AR6": Is this an accepted style for referencing the IPCC report? Please check. Should it be "First Author name YYYY"?

Thank you for pointing this out. The IPCC report citation was previously given as "IPCC WG1 AR6," and the correct citation was inadvertently omitted during sentence revision. We have corrected this to **(Masson-Delmotte et al., 2021)** in accordance with ACP reference style.

L 29–30:

"*The pace of recent climate change has intensified, prompting extensive efforts across scientific, political, and societal domains to curb its impacts (Masson-Delmotte et al., 2021).*"

L 30–32:

"*However, the nonlinear and interconnected nature of the climate system presents substantial challenges to accurately predicting outcomes and implementing effective mitigation strategies (Masson-Delmotte et al., 2021; Steffen et al., 2018).*"

L 35–37:

"*BC is a carbonaceous aerosol that is directly released into the atmosphere from incomplete combustion of biomass, fossil fuels, and biofuels. It absorbs light strongly in the visible and near-infrared spectrum (Masson-Delmotte et al., 2021).*"

**L39:** "0.14 W m$^{-2}$": Please use the minus-sign "−" instead of the hyphen "-".

We have corrected this case as well as all other instances in the manuscript where the

minus sign was typed as a hyphen, ensuring consistency throughout the text.

**L39:** You need one or more references for the BC radiative forcing "+0.14 W m$^{-2}$".

We have added the appropriate reference.

L 34–35:

*"Black Carbon (BC) is a significant SLCF that contributes roughly 0.14 W m$^{-2}$ of radiative forcing, as assessed in Masson-Delmotte et al. (2021)."*

**L45:** estimated at → estimated to be

It has been corrected.

**L47–L50:** "Historical trends prior to 2000 show … ECLIPSE inventory.": I couldn't understand this sentence. Please re-word it to be more concise.

It has been revised into two shorter sentences for clarity:

*L 44–47:*

*"BC emissions declined in North America and Europe prior to 2000, based on the Coupled Model Intercomparison Project (CMIP) inventory (Eckhardt et al., 2023). In contrast, the ECLIPSE (Evaluating the Climate and Air Quality Impact of Short-Lived Pollutants) inventory indicates a sustained increase in Asian emissions over the past two decades (Klimont et al., 2017)."*

**L68:** mass distribution → mass concentration

It has been corrected.

**L82:** "$CO/CO_2$ ratio below 0.1%": Using % for ratio (not fraction) is uncommon.

We directly calculated $CO/CO_2$ ratios as 'ppmv/ppmv × 100' and consistently report them in percent. It was initially intended to compare previous studies who reported $CO/CO_2$

ratios in % over the region including the Yellow Sea (e.g., Halliday et al., 2016). To clarify, we added the following sentence when $CO/CO_2$ is discussed in the manuscript. We believe this addition resolves potential ambiguity while allowing consistent use of % throughout the text.

L 377–378:

*"In this study, $CO/CO_2$ ratios are expressed in percent (ppmv/ppmv × 100)."*

**References:**

Halliday, H. S., Thompson, A. M., Wisthaler, A., Blake, D. R., Hornbrook, R. S., Mikoviny, T., Müller, M., Eichler, P., Apel, E. C., and Hills, A. J.: Atmospheric benzene observations from oil and gas production in the Denver-Julesburg Basin in July and August 2014, J. Geophys. Res. Atmos., 121, 11,055-011,074, https://doi.org/10.1002/2016JD025327, 2016.

Kondo, Y., Moteki, N., Oshima, N., Ohata, S., Koike, M., Shibano, Y., Takegawa, N., and Kita, K.: Effects of wet deposition on the abundance and size distribution of black carbon in East Asia, J. Geophys. Res. Atmos., 121, 4691-4712, https://doi.org/10.1002/2015JD024479, 2016.

Masson-Delmotte, V., Zhai, P., Pirani, A., Connors, S. L., Péan, C., Berger, S., Caud, N., Chen, Y., Goldfarb, L., Gomis, M. I., Huang, M., Leitzell, K., Lonnoy, E., Matthews, J. B. R., Maycock, T. K., Waterfield, T., Yelekçi, O., Yu, R., and Zhou, B. (eds.): Summary for Policymakers, in: Climate Change 2021: The Physical Science Basis. Contribution of Working Group I to the Sixth Assessment Report of the Intergovernmental Panel on Climate Change, Cambridge University Press, Cambridge, United Kingdom and New York, NY, USA, 3–32, https://doi.org/10.1017/9781009157896.001, 2021.

Moteki, N., Kondo, Y., Oshima, N., Takegawa, N., Koike, M., Kita, K., Matsui, H., and Kajino, M.: Size dependence of wet removal of black carbon aerosols during transport from the boundary layer to the free troposphere, Geophys. Res. Lett., 39, https://doi.org/10.1029/2012GL052034, 2012.

Oshima, N., Kondo, Y., Moteki, N., Takegawa, N., Koike, M., Kita, K., Matsui, H., Kajino, M., Nakamura, H., Jung, J. S., and Kim, Y. J.: Wet removal of black carbon in Asian outflow: Aerosol Radiative Forcing in East Asia (A-FORCE) aircraft campaign, J. Geophys. Res. Atmos., 117, https://doi.org/10.1029/2011JD016552, 2012.

---

## Author Comment (AC2)

**Response to Reviewer Comments**

We thank the reviewer for the constructive comments and suggestions, which helped us improve the clarity and quality of the manuscript. In this response letter, we provide point-by-point responses. Our replies to the reviewers' comments are presented **in blue font**, while the corresponding revisions in the manuscript are indicated **in blue italics with yellow highlights**, together with the **updated line numbers**.

**Comment:**

1. The star marker in Figure 1 is missing. Please ensure that all markers mentioned in the text are present in the figure.

We appreciate the reviewer for pointing this out. We have added the missing star marker in Figure 1a to indicate the starting locations of the aircraft measurements, as described in the figure caption. We also carefully checked all other figures to ensure that no additional markers are missing.

L 120:

[Figure]

**Figure 1.** (a) Flight paths for all 23 research flights conducted in this study. All flights departed from the location marked with a star. Different colors represent individual flights. The bottom-left panel shows the flight track for F220210, color-coded by altitude as a representative example of the vertical flight profiles. (b) Black carbon emission rates (tons per year / 0.1° × 0.1° grid cell) with regions where air masses influencing the observations during the aircraft mission (South Korea, North Korea, Japan, China, Mongolia, Russia, and LRT in different colors). BC emissions are sourced and averaged from EDGARv8.1 (Crippa et al., 2024) data for 2021 and 2022.

2. Line 62: A recent study (Hu et al., EST Letters, 2022, 10.1021/acs.estlett.2c00060) identified the fraction of more spherical BC which can support your statements about the influence of particle morphology on BC absorption properties.

Thank you for this valuable suggestion. We have incorporated the reference and expanded the discussion accordingly:

L 58–61:
*"Recently, Hu et al. (2022) reported that freshly emitted or only slightly coated BC generally exhibits a fractal-like structure, becoming progressively more spherical as coatings fully envelop the core. This gradual change toward sphericity is closely linked to the efficiency of the lensing effect that enhances BC light absorption."*

3. Line 89: The aircraft measurements about BC over China were also conducted in recent studies (Hu et al, Chemosphere, 2020, 1016/j.chemosphere.2020.126455; Tian et al., ACP, 2020, 10.5194/acp-20-2603-2020; Liu et al., ERL, 2019, 10.1088/1748-9326/ab4872) could be cited to discuss aircraft measurement experiments conducted over the Chinese megacities of Beijing and Xuzhou.

We agree with the reviewer's suggestion and have added these references both in the Introduction and in the Results and Discussion sections:

L 93–95:
*"Most recent aircraft studies over Asia have focused on observations conducted primarily over inland urban areas including Chinese megacities such as Beijing and Xuzhou (Hu et al., 2020; Liu et al., 2019a; Tian et al., 2020), with limited coverage of marine or downwind regions."*

L 366–369:
*"Similar aloft-enhanced processes were also reported by Liu et al. (2019a), reported enhanced secondary aerosol formation and BC coating above the PBL under strong midday solar radiation, emphasizing the role of upper-atmosphere photochemistry in driving BC aging under intense sunlight."*

4. Line193-195: Please provide a justification for choosing a height of 2.5 km instead of other

heights.

Since BC is emitted near the surface, our aircraft observations show that rBC mass concentrations sharply decline above ~3 km, likely suggesting that vertical mixing processes can effectively loft surface-emitted BC to altitudes approaching 3 km before significant dilution or removal occurs.

To focus on the lower troposphere where surface emissions remain influential, we selected 2.5 km as a conservative upper bound for analysis. This threshold is supported by previous studies indicating that land-based PBLH can reach up to ~2 km under favorable convective conditions (Gu et al., 2020; Qu et al., 2017). Furthermore, several regional aerosol transport studies have adopted 2.5 km as a practical reference altitude (Kanaya et al., 2013; Kanaya et al., 2016; Choi et al., 2020), while Lamb et al. (2018) used model-derived mixing depths along trajectories for similar purposes.

The manuscript text has been revised accordingly to incorporate this clarification.

L 202–205:
*"Following previous studies (Choi et al., 2020b; Kanaya et al., 2013, 2016), the country most frequently traversed by each trajectory below 2.5 km altitude was designated as the origin of the air mass. This threshold reflects the fact that, under favorable convective conditions, the PBL over land can extend up to ~2 km (Gu et al., 2020; Qu et al., 2017), allowing surface-emitted aerosols to reach at least this altitude."*

5.  Line 187: Clarify whether the 10s indicates outputting one trajectory every 10 seconds or if it represents the time step for one trajectory. If it is the output frequency, explain how it is implemented. If it is the time step, note that the minimum time step in HYSPLIT is 1 minute, as mentioned in the HYSPLIT Limitations. Additionally, provide information about your target site.

The '10s' refers to the interval of our observational data, not to the HYSPLIT model time step. For each 10 seconds observation along aircraft flight tract (latitude, longitude, and altitude), we computed a back trajectory using HYSPLIT model. Because HYSPLIT requires meteorological input at hourly resolution, the 10 seconds observation times were rounded to the nearest hour

when they supplied to the model. We have revised the manuscript to clarify this point.

The information on our target site, Yellow Sea is written as follows.

L 116–118:

"The YS, located west of the Korean Peninsula and downwind of continental East Asia, serves as an ideal receptor site for observing air masses transported from continental regions toward the Korean Peninsula under the prevailing westerlies."

In our HYSPLIT modeling, backward air mass trajectories were calculated at each observational point over the Yellow Sea, it is clearly indicated in the following sentence.

L 195–197:

"At 10-second intervals along the flight track over the YS, 5-day back trajectories were computed using the Global Data Assimilation System (GDAS1, 1 ˚× 1 ˚ resolution), consistent with BC's atmospheric lifetime. "

6.  The different types of pollution in Figure 2 are interesting. However, the authors did not conduct further analysis and research on this. Please explain the reason for classifying pollution types and how it contributes to the study's objectives.

Earlier versions of the manuscript included more detailed analysis of haze and dust events; these were later streamlined to focus on the main objectives. Nevertheless, we retained the indication of pollution events in Figure 2 because our multi-year aircraft measurements inevitably sampled a variety of atmospheric conditions. Marking Asian Dust and Haze episodes helps readers interpret day-to-day variations in concentration and mixing state and provides useful context for other researchers interested in these events over the Yellow Sea. We have added corresponding content to the manuscript.

L 271–274:

"In addition, we captured several episodic events including "Haze", "Asian Dust", and the mixed "Haze & Asian Dust", as classified by Korea Meteorological Administration (KMA). During Asian Dust episodes, most variables remained comparable to the other days, except for a noticeable decrease in BC's internal mixing. In contrast, Haze events were characterized by substantial

*increases in the average values of all observed parameters.* Thus, both size distributions (MMD) and mixing state ($F_{thick}$ and $R_{shell/core}$) of rBC particles observed in this study clearly indicate their considerable dependence on the origins and further chemical/physical processes of the air masses during transport to this remote environment."

7. Figure 4: The figure suggests a regional transmission process above 3000 m in autumn, leading to changes in BC mass and MMD. However, this result is not reflected in the coating thickness, which appears to maintain the same trend as at low altitudes. Please provide an explanation for this observation.

We appreciate this insightful comment. For more detailed analysis and discussion, we have added two figures in SI (Figure S7 and Figure S8).

Prior to discussing the autumn observations in detail, we briefly address the complexity of rBC mixing state characteristics identified in our measurements. Our data analysis demonstrates the observed BC's internal mixing over the Yellow Sea is primarily determined by the source origin of air masses. Table 2 summarizes the average rBC properties by major source region ('Major Region') identified using backward air mass trajectories. The majority of air masses originated from China (62 %), which exhibited the high degree of rBC internal mixing, whereas those classified as Long-Range Transport (LRT) showing the lowest. This airmass origin-varying physical properties of rBC particles influenced their vertical profiles (Figure 4). In winter, Chinese air masses dominated below 2.5 km (74 %), whereas above this altitude, the LRT contribution rises to 88 % (Figure S7), accompanied by sharp decreases in both $F_{thick}$ and $R_{shell/core}$ (Figure 4). In contrast, during spring, together with potentially enhanced photochemical reaction at upper altitude, the contrast in source regions was less pronounced, leading to more moderate vertical gradients in mixing-state parameters.

Regarding the autumn observations, figure S7 shows the vertical profiles of rBC properties and accumulated precipitation along trajectories (APT) for each daily observation during the autumn. The autumn campaign included three aircraft flight days. Among them, two flights (F211001 and F211102) exhibited very similar vertical profiles of rBC properties and APT. Meanwhile, during F211125, a sharp decrease in $M_{rBC}$ and MMD was observed above ~2 km, coinciding with an increase in APT to ~2–6 mm. This alignment suggests that rBC particles, particularly larger ones, were scavenged by precipitation, leading to the observed reductions in $M_{rBC}$ and MMD. This

episodic case (F211125) influenced the average shape of the autumn vertical profile (Figure 4).

However, two mixing-state parameters ($F_{thick}$ and $R_{shell/core}$) did not follow the same vertical patterns as $M_{rBC}$ and MMD. Compared to the other seasons, both $F_{thick}$ and $R_{shell/core}$ remained consistently high with altitude on all three autumn flight days, likely due to the combined effects of (i) enhanced photochemical production of coating materials and shorter transport distance than winter and (ii) the relatively low solubility of those coatings, etc. Meanwhile, the vertical gradient of $R_{shell/core}$ was steeper than that of $F_{thick}$. This can be attributed largely to their size-selection difference in methods: $F_{thick}$ is determined from all detected rBC particles (~70–510 nm), whereas $R_{shell/core}$ is derived from rBC cores in the 140–220 nm range. Preferential wet scavenging of larger rBC core (and/or soluble coating) at upper altitude likely contributed to the steeper decline of $R_{shell/core}$ with altitude at some extent.

These findings demonstrate that the rBC mixing state is primarily determined by combustion activity and conditions at the source, but is further modified significantly by atmospheric processes such as wet scavenging and photochemical reaction, ultimately shaping its seasonal vertical structure. These discussions have been added as follows.

[Figure]

**Figure S7.** Vertical profiles of rBC particles ($M_{rBC}$, MMD, $F_{thick}$, $R_{shell/core}$, APT) for 2021 Autumn (Oct.–Nov.), F211001, F211102 and F211125. Horizontal error bars indicate ± 1 σ (standard deviation) variability within each altitude bin.

[Figure]

**Figure S8.** Seasonally varying vertical contributions of Major Regions for Winter, Spring, and Autumn.

L 344–359:

"*In line with these studies, our findings suggest that aged rBC particles transported to high altitudes (above ~3 km) undergo significant physical transformation largely due to wet scavenging processes such as precipitation and cloud interaction, highlighting the strong sensitivity of both their size and mixing state to these removal mechanisms.*

*As a case, during F211125 in the autumn, a sharp decrease in $M_{rBC}$ and MMD was observed above ~2 km, coinciding with an increase in APT to ~2–6 mm (Figure S7). This alignment suggests that rBC particles, particularly larger ones, were scavenged by precipitation, leading to the observed reductions in $M_{rBC}$ and MMD. This episodic case (F211125) influenced the average shape of the autumn vertical profile (Figure 4). However, two mixing-state parameters ($F_{thick}$ and $R_{shell/core}$) did not follow the same vertical patterns as $M_{rBC}$ and MMD. Compared to the other seasons, both $F_{thick}$ and $R_{shell/core}$ remained consistently high with altitude on all three autumn flight days (Figure S7), likely due to the combined effects of (i) enhanced photochemical production of coating materials and shorter transport distance than winter and (ii) the relatively low solubility of those coatings, etc. Meanwhile, the vertical gradient of $R_{shell/core}$ was steeper than that of $F_{thick}$. This can be attributed to their size-selection difference in methods: $F_{thick}$ is determined from all detected rBC particles (~70–510 nm), whereas $R_{shell/core}$ is derived from rBC cores in the 140–220 nm range. Preferential wet scavenging of larger particles at upper altitude likely contributed to the steeper decline of $R_{shell/core}$ with altitude at some extent.*"

L 427–434

"This airmass origin-varying physical properties of rBC particles influenced their vertical profiles (Figure 4). In winter, Chinese air masses dominated below 2.5 km (74 %), whereas above this altitude, the LRT contribution rises to 88 % (Figure S7), accompanied by sharp decreases in both $F_{thick}$ and $R_{shell/core}$ (Figure 4). In contrast, during spring, together with potentially enhanced photochemical reaction at upper altitude, the contrast in source regions was less pronounced, leading to more moderate vertical gradients in mixing-state parameters. These findings provide clear evidence that the vertical structure of rBC mixing state is fundamentally shaped by combustion characteristics at the source region and thus vary seasonally with changes in air mass origin. This structure is further modified by atmospheric processes such as wet scavenging and photochemical aging.

L504 in Conclusion:

"Our observational results from YS provide the clear evidence that the vertical structure of rBC's mixing state was fundamentally shaped by combustion characteristics at the source region and further modified substantially by atmospheric processes such as wet scavenging and photochemical aging."

8. Figure 5a: The transport processes of gases and aerosols differ below and above the boundary layer. Explain how the authors justify using the same fitting result to represent the results at different heights. Additionally, discuss the reason for Russia's significantly lower r2 compared to other regions.

We acknowledge that transport processes differ between the planetary boundary layer (PBL) and the free troposphere (FT).

In an earlier version of the analysis, we applied separate fits for data within the PBL and in the FT, but except for a few cases such as Russia, the slopes and correlations were nearly identical to those obtained with a single fit. As shown in Figure 5a, the data points from the PBL and FT are distinguished by different markers, yet the relationship between $M_{rBC}$ and CO is well described by essentially the same slope, which justified using a single fitting line to represent both altitude ranges.

Regarding the relatively low $R^2$ for Russia (0.34), several factors may contribute:

- Source heterogeneity: High $R^2$ values primarily imply that $M_{rBC}$ and CO originate from similar combustion sources. A lower $R^2$ suggests that the two species may have different emission sources or that the sampled air masses were influenced by multiple plumes during transport. Russia covers a very large geographical area, and emissions differ substantially between its western and eastern regions. Such heterogeneity may reduce the correlation.

- Trajectory classification uncertainty: When we separate Russian air masses by altitude, the $R^2$ in the FT improves to 0.68, while it remains low (0.18) in the PBL. Air masses entering the Yellow Sea from Russia generally travel at higher altitudes; those sampled in the PBL are more likely to have interacted with plumes from Mongolia or China on route,

which can weaken the $M_{rBC}$–CO correlation.

These combined factors plausibly explain the lower $R^2$ for Russia compared with other source regions. We have added this discussion to the manuscript.

L 443–449:

*"In addition, Russia-sourced air exhibited notable vertical contrasts in both tracer ratios and rBC-CO correlation. While the overall rBC-CO correlation was relatively weak ($R^2$ = 0.34; Figure 5), this likely reflects the heterogeneous emission sources across the vast Russian territory and the influence of mixed plumes during transport. However, when separated by altitude, the correlation improved significantly in the FT ($R^2$ = 0.68) but remained low in the PBL ($R^2$ = 0.18), suggesting greater mixing with other continental plumes at lower altitudes. This vertical pattern was further supported by lower rBC/CO and higher $CO/CO_2$ slopes in the PBL, likely influenced by additional CO emissions along the transport pathway, while the opposite was observed in the FT."*

**References:**

Choi, Y., Kanaya, Y., Park, S. M., Matsuki, A., Sadanaga, Y., Kim, S. W., Uno, I., Pan, X., Lee, M., Kim, H., and Jung, D. H.: Regional variability in black carbon and carbon monoxide ratio from long-term observations over East Asia: assessment of representativeness for black carbon (BC) and carbon monoxide (CO) emission inventories, Atmos. Chem. Phys., 20, 83–98, https://doi.org/10.5194/acp-20-83-2020, 2020b.

Gu, J., Zhang, Y., Yang, N., and Wang, R.: Diurnal variability of the planetary boundary layer height estimated from radiosonde data, Earth Planet. Phys., 4, 479–492, https://doi.org/10.26464/epp2020042, 2020.

Hu, K., Zhao, D., Liu, D., Ding, S., Tian, P., Yu, C., Zhou, W., Huang, M., and Ding, D.: Estimating radiative impacts of black carbon associated with mixing state in the lower atmosphere over the northern North China Plain, Chemosphere, 252, 126455, https://doi.org/10.1016/j.chemosphere.2020.126455, 2020.

Hu, K., Liu, D., Tian, P., Wu, Y., Li, S., Zhao, D., Li, R., Sheng, J., Huang, M., Ding, D., Liu, Q., Jiang, X., Li, Q., and Tao, J.: Identifying the fraction of core–shell black carbon particles in a complex mixture to constrain the absorption enhancement by coatings, Environ. Sci. Technol. Lett., 9, 272–279, https://doi.org/10.1021/acs.estlett.2c00060, 2022.

Liu, D., Zhao, D., Xie, Z., Yu, C., Chen, Y., Tian, P., Ding, S., Hu, K., Lowe, D., Liu, Q., Zhou, W., Wang, F., Sheng, J., Kong, S., Hu, D., Wang, Z., Huang, M., and Ding, D.: Enhanced heating rate of black carbon above the planetary boundary layer over megacities in summertime, Environ. Res. Lett., 14, 124003, https://doi.org/10.1088/1748-9326/ab4872, 2019.

Qu, Y., Han, Y., Wu, Y., Gao, P., and Wang, T.: Study of PBLH and its correlation with particulate matter from one-year observation over Nanjing, Southeast China, Remote Sens., 9, 668, https://doi.org/10.3390/rs9070668, 2017.

Tian, P., Liu, D., Zhao, D., Yu, C., Liu, Q., Huang, M., Deng, Z., Ran, L., Wu, Y., Ding, S., Hu, K., Zhao, G., Zhao, C., and Ding, D.: In situ vertical characteristics of optical properties and heating rates of aerosol over Beijing, Atmos. Chem. Phys., 20, 2603–2622, https://doi.org/10.5194/acp-20-2603-2020, 2020.